# Effects of Different Combinations of Sodium Butyrate, Medium-Chain Fatty Acids and Omega-3 Polyunsaturated Fatty Acids on the Reproductive Performance of Sows and Biochemical Parameters, Oxidative Status and Intestinal Health of Their Offspring

**DOI:** 10.3390/ani13061093

**Published:** 2023-03-19

**Authors:** Caiyun You, Qingqing Xu, Jinchao Chen, Yetong Xu, Jiaman Pang, Xie Peng, Zhiru Tang, Weizhong Sun, Zhihong Sun

**Affiliations:** Key Laboratory for Bio-Feed and Animal Nutrition, Southwest University, Chongqing 400715, China

**Keywords:** fatty acids, antioxidant capacity, intestinal microflora, sows, piglets

## Abstract

**Simple Summary:**

Dietary supplementation with fatty acids benefits the high productivity of sows and plays an essential role in piglet growth. Considering that the mechanisms of the fatty acid types on animal physiology differ, combined supplementation may have additive effects. Therefore, we investigated the effects of different combinations of fatty acids with different chain lengths on the reproductive performances of sows and on the antioxidant capacity, immune function, and intestinal health of their offspring. Taken together, the dietary supplementation of sows with different combinations of SB, MCFAs, and omega-3 PUFAs to the sows during late gestation and lactation can efficiently improve the growth performance, immune function, antioxidant capability, and intestinal microbiota and decrease the incidence of diarrhea in the suckling piglets. Additionally, dietary SMP supplementation has better effects on piglet intestinal health and is likely through gut microorganism alterations.

**Abstract:**

The aim of the study was to investigate the comparative effects of different combinations of sodium butyrate (SB), medium-chain fatty acids (MCFAs), and omega-3 polyunsaturated fatty acids (n-3 PUFAs) on the reproductive performances of sows, as well as on the biochemical parameters, oxidative statuses, and intestinal health of the sucking piglets. A total of 30 sows were randomly allocated to five treatments: (1) control diet (CON); (2) CON with 1 g/kg of coated SB and 7.75 g/kg of coated MCFAs (SM); (3) CON with 1 g/kg of coated SB and 68.2 g/kg of coated n-3 PUFAs (SP); (4) CON with 7.75 g/kg of coated MCFAs and 68.2 g/kg of coated n-3 PUFAs (MP); (5) CON with 1 g/kg of coated SB, 7.75 g/kg of coated MCFAs and 68.2 g/kg of coated n-3 PUFA (SMP). The results showed that sows fed the SP, MP, and SMP diets had shorter weaning-to-estrus intervals than those fed the CON diet (*p* < 0.01). The piglets in the SM, SP, and MP groups showed higher increases in the plasma catalase and glutathione peroxidase activities than those of the CON group (*p* < 0.01). The diarrhea incidence of piglets in the SM, SP and SMP groups was lower than that of piglets in the CON group (*p* < 0.01). Additionally, the addition of SM, SP, MP, and SMP to the sow diets increased the contents of immunoglobulin A, immunoglobulin G, fat, and proteins in the colostrum (*p* < 0.01), as well as the plasma total superoxide dismutase activities (*p* < 0.01) in the suckling piglets, whereas it decreased the mRNA expressions of tumor necrosis factor-α, interleukin-1β, and toll-like receptor 4 in the jejunum mucosa of the piglets. The relative abundances of *Prevotella*, *Coprococcus*, and *Blautia* in the colonic digesta of the piglets were increased in the SM group (*p* < 0.05), and the relative abundances of *Faecalibacterium* increased in the SMP group (*p* < 0.05), compared with the CON group. The relative abundances of *Collinsella*, *Blautia*, and *Bulleidia* in the MP group were higher than those in the CON group (*p* < 0.05). Collectively, dietary combinations of fatty acids with different chain lengths have positive effects on the growth performances and intestinal health of suckling piglets.

## 1. Introduction

The reproductive performances of high-yielding sows and the growth performances of piglets are the two most important aspects that influence the economic efficiency of the modern pig breeding industry. With the development of breeding techniques, the reproductive performances of sows have been improved. However, highly prolific sows often suffer from a range of issues, including insufficient nutrient intake, excessive weight loss, longer weaning-to-estrus intervals (WEI), shortened service lives, and metabolic disorders, which result in the retarded growth of their piglets [1,2,3,4]. Additionally, in late pregnancy, the fetal growth rate dramatically accelerates [5]. Diets that contain supplemental fatty acids have been particularly effective at improving the body conditions of sows and the birth weights and growth of suckling piglets [6,7,8]. Therefore, meeting the nutrient requirements of prolific sows is a vital consideration for the development of pig farming.

Fatty acids have biological functions, such as the regulation of metabolic disorders, the intestinal barrier, and the immune function of animals [7,9,10]. Fatty acids are categorized according to the length of the carbon chain and the degree of saturation. Short-chain fatty acids (SCFAs), and especially butyrate (SB), which is a major source of energy for colonic epithelial cells [9], can improve the immune function of piglets through the nuclear factor-kappa B (NF-κB) signal pathway [11,12]. In addition, appropriate doses of butyrate can alleviate diarrhea symptoms and reduce the intestinal permeability to maintain intestinal health [13]. The majority of MCFAs are carried straight to the liver via the portal vein, granting a rapid energy supply, which is vital for piglets [7]. In addition, MCFAs decrease intestinal colonization by opportunistic pathogens and modulate the colonic microbiota of piglets [14,15]. n-3 PUFAs are essential for embryonic and fetal development [16]. In addition, n-3 PUFAs also play an important role in shortening the weaning-to-estrus interval (WEI) and enhancing the immune function in sows [16,17,18]. Previous studies have shown that dietary single fatty acid supplementation could shorten the WEIs of sows and improve the intestinal health and growth performances of suckling piglets [19,20]. However, studies on the effects of dietary combinations of SB, MCFAs, and n-3 PUFA on prolific sows during late gestation and lactation are still lacking. Therefore, the purpose of this study was to investigate the effects of different combinations of SB, MCFAs, and n-3 PUFAs on the reproductive performances of sows and on the biochemical parameters, oxidative status, and intestinal health of their offspring during late gestation and lactation. We speculated that the combination of fatty acids would have additive effects on the reproductive performances of the sows and the growth performances of their offspring.

## 2. Materials and Methods

All animal operations were carried out in compliance with protocols approved by the Animal Ethics Committee of Southwestern University (Chongqing, China). The behavior and health of the experimental animals were continuously monitored during the trial period, and no negative impacts were observed. More precisely, there were no clinical problems that could have necessitated pharmaceutical treatment for pathologies after the investigation began, and all the animals were deemed suitable for the study. The experimental animals were disposed of safely following the Experimental Animal Handling Procedure of Southwest University (Ethics Approval Code: IACUC-20210120-03).

### 2.1. Animals, Materials, and Feeding Management

A total of 30 third-parity sows (Landrace × Large White hybrid; 200 ± 15 kg) were used in this study. From mating to d 109 of gestation, the sows were kept in individual stainless-steel cages (0.60 × 2.15 m) in the gestation house, and on approximately d 110 of gestation, they were transferred to the farrowing stalls (1.20 × 2.15 m) in a thoroughly sterilized farrowing house with iron fencing and plastic flooring. The SB, MCFAs, and omega-3 PUFAs in this study were supplied by Xingao Agribusiness Development Co., Ltd. (Xiamen, Fujian, China), with purities of 98%, 70%, and 20%, respectively. The primary active constituents of the n-3 PUFAs were docosahexaenoic acid (DHA), α-linolenic acid (ALA), and eicosapentaenoic acid (EPA). Additionally, the sows were fed twice a day at 08:00 and 16:00, with 2.5–3.0 kg of feed per day, which was restricted in late gestation based on the body condition, while 2 kg was fed on d 1–2 of lactation, with an increase of 0.5 kg per day from d 3 to d 7 of lactation, and an increase of 0.8 kg per day from d 8 to d 14 of lactation, with no further increase thereafter. All sows were allowed to consume water at any time during the study. Heaters and exhaust fans kept the room at a comfortable temperature (from 20 to 25 °C).

### 2.2. Diets and Experimental Design

The sows were randomly allocated to five treatments (six replicate pens per treatment and one sow per replicate) in a completely randomized experimental design. The sows were fed a basal diet (control, CON), a basal diet supplemented with 1 g/kg of coated SB and 7.75 g/kg of coated MCFAs (SM), a basal diet supplemented with 1 g/kg of coated SB and 68.2 g/kg of coated n-3 PUFAs (SP), a basal diet supplemented with 7.75 g/kg of coated MCFAs and 68.2 g/kg of coated n-3 PUFAs (MP), and a basal diet supplemented with 1 g/kg of coated SB, 7.75 g/kg of coated MCFAs, and 68.2 g/kg of coated n-3 PUFAs (SMP). The dosages were chosen based on the company’s recommended dosages. The piglets were housed in farrowing stalls, with one litter per pen. After the piglets were born, they were manually attached to the nipples to guarantee that they received breast milk for growth and development. This study shared the control group with Chen et al. [20], and the sows in this trial and Chen’s sows were kept in the same barn. The trial started on d 85 of gestation and ended with the weaning of the piglets (d 21 of lactation). From d 85 to d 110 of gestation, the test sows were fed given the gestation diet, followed by the lactation diet from d 110 of gestation and throughout weaning. The nutritional content of the baseline diet met or surpassed the nutritional recommendations of the National Research Council (2012) [21]. The dietary ingredients and nutritional levels for the sows throughout gestation and lactation are shown in Table 1. The gestation diet (approximately 100 g) and lactation diet (approximately 100 g) were collected. Then, the feed samples were analyzed for the crude protein (CP), crude ash (Ash), dry matter (DM), ether extract (EE), calcium (Ca), crude fiber (CF), available phosphorus (AP), and total phosphorus (Total P), according to the procedures followed by the standard of the AOAC (2000) [22].

### 2.3. Recording and Sample Collection

#### 2.3.1. Reproductive Performances of Sows

During the animal experiment, the feed wastage was recorded every day after the meal to calculate the average daily feed intake (ADFI). The individual neonatal weight of the born alive was weighed within 12 h of delivery. The stillborn and mummified fetuses were not weighed, and the mummified fetuses were counted as stillbirths (piglets that died before birth). We measured the numbers of born alive, stillborn, mummified fetuses, and total born, the litter weights, birth weights, and the weaning weights of the piglets, and the WEIs of the sows. The total litter size included mummified fetuses, born alive, and stillborn. The born-alive rate was calculated as the number of born alive/total born × 100%.

#### 2.3.2. Growth Performances of Piglets and Diarrhea Incidence

Piglets were cross-fostered after altering the litter sizes of the sows within the same treatment within 24 h of farrowing. Piglets were weaned on d 21 of lactation, the number of weaned piglets was counted, and the piglets were weighed to determine the ADGs and weaning survival rates. The diarrhea severity was determined by daily observation of the piglet feces, as previously described [23,24]. The formula for calculating the diarrhea incidence was as follows: the diarrhea rate (%) = Σ[(diarrhea days in piglets × number of diarrhea piglets)]/(total number of piglets × 21) × 100%.

#### 2.3.3. Sample Collection

The colostrum (about 40 mL) was manually collected after the alcohol sterilization of the sow teats within 2 h of the first piglet’s birth. For each repetition, piglets that met the average weight were randomly selected for blood and slaughter sampling. On d 22 of lactation, one piglet per pen was randomly selected for the collection of blood samples from the anterior vein. Then, the blood samples were centrifuged (4 °C, 3000× *g*, 15 min) and the plasma was stored at −80 °C for subsequent analysis. Next, the piglets were humanely killed after anesthesia by intravenous injection with sodium pentobarbital (50 mg/kg BW). Tissue samples from the liver and middle jejunum were taken, flushed with 0.9% saline, and deposited in a 4% formaldehyde solution for morphological examination. The colonic digesta (approximately 10 g) was collected in sterile tubes for the microbiota analysis. After that, jejunal mucosa samples were carefully scraped off using a sterile glass slide, flash-frozen in liquid nitrogen and were maintained at −80 °C for subsequent analysis.

### 2.4. Analytical Methods

#### 2.4.1. Colostrum Composition Analysis

One colostrum sample (approximately 20 mL) diluted 3 times with purified water was determined in triplicate for milk fat, milk protein, lactose and solids-not-fat (SNF) using a FOSS Multifunctional Dairy Analyzer (MilkoScan TM FT120, Foss Electric A/S, Hillerød, Denmark). The other colostrum sample (approximately 20 mL) was centrifuged (4 °C, 3000× *g*, 20 min), and the supernatant was aspirated and kept at −80 °C. The colostrum supernatant was thawed at room temperature and utilized to determine the concentrations of immunoglobulin A (IgA), immunoglobulin G (IgG), and immunoglobulin M (IgM) using swine reagent ELISA kits provided by the Nanjing Jiancheng Bioengineering Institute (Nanjing, Jiangsu, China). The intra- and inter-assay CVs for these ELISA kits were both less than 9.0%.

#### 2.4.2. Blood Biochemical Parameters

The total antioxidant capacity (T-AOC); (Code: A015-2-1), glutathione peroxidase (GSH-Px); (Code: A005-1-1), malondialdehyde (MDA); (Code: A003-1-2), total superoxide dismutase (T-SOD); (Code: A001-1-1) and catalase (CAT); (Code: A007-1-1) in the plasma of the pigs were analyzed by using commercial assay kits (Nanjing Jiancheng Bioengineering Institute, Nanjing, Jiangsu, China). The concentrations of total protein (TP); (Code: A045-1-1) and albumin (ALB); (Code: A028-2-1) in the plasma of the pigs were determined using a por-cine-specific commercial kit with microplate test methods and an enzyme-labeled instrument (Thermo Electron Corporation; Rochester, NY, USA). The plasma concentrations of high-density lipoprotein cholesterol (HDL-C); (Code: A112-1-1), glucose (GLU); (Code: F006-1-1), total cholesterol (TC); (Code: A111-1-1), urinary nitrogen, and triglycerides (TG); (Code: A110-1-1) were determined using colorimetric method diagnostic kits. In accordance with the manufacturer guidelines, all protocols were strictly carried out, ensuring the highest level of safety and accuracy.

#### 2.4.3. Intestinal Morphology

Jejunum tissues were collected from weaning piglets and fixed in 4% formalin for the analysis of the intestinal morphology. In short, the villus height (VH) and crypt depth (CD) of the jejunum were measured using an Axio Scope A1 microscope (Zeiss, Oberkochen, Germany) with 40× combined magnification. The VH was calculated by measuring the distance between the top of the villus and the villus-crypt junction, and the CD was calculated by measuring the distance between the villus-crypt junction and the bottom of the crypt. The averages of the measurements (at least 10 normative measurements) were used for the statistical analysis, and all intestinal mucosal morphometric analyses were executed by the same operator.

#### 2.4.4. Quantitative Real-Time PCR

The relative expressions of the genes related to the factors involved in the regulation of inflammation in the jejunal mucosa of the piglets were determined. Total RNA was isolated from frozen jejunal mucosa samples by using SteadyPure Uni-versal RNA Extraction Kits II (Code: AG21022; Accurate Biotechnology (Hunan) Co., Ltd., Changsha, China). The specific RNA extraction procedure was performed using the manufacturer’s recommendations. The concentration of total RNA was measured with a NanoDrop-ND2000 spectrophotometer (ThermoFisher Scientific, Waltham, MA, USA), and the reverse transcription was performed with the qualified RNA samples using AMV First Strand cDNA Synthesis Kits provided by Sangon (Shanghai, China), according to the manufacturer’s instructions.

Real-time PCR analysis was performed to quantify the claudin-*1* (*CLDN 1*), interleukin-*6* (*IL*-*6*), tumor necrosis factor-α (*TNF-α*), occludin (*OCLN*), interleukin-1β (*IL*-1*β*), zonula occludens-*1 (ZO*-*1*), interleukin-10 (*IL*-10), toll-like receptor 4 (*TLR*-*4*), *NF-κB*, myeloid differentiation factor 88 (*MγD88*), and glyceraldehyde-3-phosphate (*GAPDH*) mRNA levels in the jejunal mucosa. The primer sequences for all the target genes and predicted product sizes are shown in Table 2. The real-time PCR analysis was conducted using the SYBR Green approach combined with an ABI 7900 Sequence Detection System. The following thermal cycling parameters were used: initial denaturation at 94 °C for 30 s, followed by 40 cycles at 94 °C for 5 s, annealing temperature for 20 s, and extension at 72 °C for 20 s. Moreover, the melting curve analysis was used to ensure that the PCR products remained specific and pure. A standard curve was generated using LightCycler software and the amplification of serially diluted cDNA, and the quantification of the target gene expression was calculated using the 2^−ΔΔCT^ method based on the standard curve with the *GAPDH* gene as the reference gene [25].

#### 2.4.5. 16S rRNA Gene Sequencing and Microbiota Analysis

Total DNA from the digesta of the colon was extracted using Power Fecal DNA Isolation Kits (Mobio, Carlsbad, CA, USA) following the manufacturer’s instructions. Briefly, the 16S rDNA gene was presented in the genome of all bacteria and was highly conserved. Microbial profiling was performed on an Illumina HiSeq2500 platform (Novogene, Beijing, China) by the PCR amplification of a segment of a highly variable region sequence (V3 region) following 600 amplification cycles. Then, the raw data sequences for the 16S rRNA gene were collected and filtered with the software tools FLASH and QIIME. UPARSE was used to assess the valid sequences and establish the practical classification units (OTUs). Moreover, singletons and OTUs below 0.005% were eliminated. Subsequently, the species-level classification was determined by the taxonomic alignment of high-quality sequences with the National Center for Biotechnology Information (NCBI) nucleotide database (ver. 2.20) at a 90% confidence threshold. The alpha diversity index (Shannon, ACE, Chao1, and Simpson) and β-diversity (Bray Curtis) analyses were calculated with the QIIME software tool.

### 2.5. Statistical Analysis

According to the post hoc power analysis for the ADFI of the sows, ADGs of the piglets, and IgA in the colostrum and plasma in this study, the calculated statistical power was > 0.90; thus, 6 pigs per treatment were enough to provide sufficient statistical power (α < 0.05; β = 0.80). All data analyses were performed using ANOVA analysis, with the dietary treatments used as the fixed factor. Diarrhea rate data assessments were translated using the arcsine square root transformation for subsequent statistics. Data were subjected to Duncan’s test method using SPSS 19.0 software (SPSS Inc., Chicago, IL, USA), and they are presented as means and standard errors of means (SEMs) unless otherwise noted. The histograms were created using GraphPad Prism 8. (GraphPad Prism Inc., San Diego, CA, USA). Statistical significance was identified when *p* < 0.05, and trends were considered when 0.05 < *p* ≤ 0.10.

## 3. Results

### 3.1. Reproductive Performances of Sows

As shown in Table 3, there were no significant changes in the number of born alive, stillborn and total born among the dietary treatments (*p* > 0.05). Compared with the CON group, the ADFI of the sows was significantly increased in the SM and SMP groups (*p* < 0.01). Moreover, the sows fed the SP, MP, and SMP diets showed significantly shorter WEIs compared with those fed the CON diet (*p* < 0.01).

### 3.2. Growth Performances of Piglets

As shown in Table 3, there were no differences in the survival rates of the suckling piglets among the dietary treatments (*p* > 0.05). The final BWs of the piglets in the SMP group were significantly higher than those of the piglets in the CON, SM, and MP groups (*p* < 0.01). The ADGs of the piglets in the SMP group were higher than those of the suckling piglets in the CON and MP groups (*p* < 0.01). Of note, the suckling piglets in the SMP group showed higher final BWs and ADGs than the other groups. Moreover, the diarrhea incidence of the suckling piglets in the CON group was higher than those of the piglets in the SM, SP, and SMP groups (*p* < 0.01).

### 3.3. Colostrum Composition of Sows

As shown in Table 4, compared with the sows fed the control diet, the dietary addition of SM, SP, MP, and SMP increased the concentrations of fat and protein in the colostrum (*p* < 0.01). The SNF concentrations in the colostrum of the sows in the SM and MP groups were higher than those of the sows in the CON, SP, and SMP groups (*p* < 0.01). In addition, the concentrations of IgA, IgG, and IgM in the colostrum of the SM, SP, and SMP groups were higher than those of the sows in the CON group (*p* < 0.01).

### 3.4. Plasma Biochemical Index of Suckling Piglets

As shown in Table 5, the plasma TP, FFA, and HDL contents of the suckling piglets were significantly increased in the SM, SP, MP, and SMP groups compared with those of the piglets in the CON group (*p* < 0.01). In addition, the TG and TC contents were decreased in the plasma of the suckling piglets in the SM, MP, and SMP groups (*p* < 0.01) compared with those in the plasma of the piglets in the CON group. For the immunoglobulin levels, the piglets in the SM group showed higher IgA concentrations in their plasma than the other groups (*p* < 0.001). Compared with the CON group, the SM, SP, MP, and SMP groups showed significantly increased IgG concentrations in the plasma of the piglets (*p* < 0.01).

### 3.5. Plasma Antioxidant Capacity of Suckling Piglets

As shown in Table 6, the piglets in the SM, SP, MP, and SMP groups showed significantly increased plasma T-SOD and T-AOC activities in comparison with those in the CON group (*p* < 0.01). The piglets in the SM, SP, and MP groups had higher plasma CAT and GSH-Px activities than those in the CON group (*p* < 0.01), and the piglets in the MP group had the highest plasma GSH-Px activity among the five groups. However, the content of plasma MDA was higher in the MP group than in the CON group (*p* < 0.01), with no significant difference in the SM, SP, and SMP groups (*p* > 0.05).

### 3.6. Intestinal Morphology of Sucking Piglets

As shown in Figure 1B, there was no significant difference in the VH of the jejunum among the five groups (*p* > 0.05). The CD of the jejunum in the SM, SP, MP, and SMP groups was significantly lower than that of the jejunum in the CON group (*p* < 0.01, Figure 1C). The VH/CD ratio of the jejunum in the SM, SP, MP, and SMP groups was significantly increased compared with that of the CON group (*p* < 0.01, Figure 1D). Moreover, the piglets in the SMP group showed a higher VH/CD ratio in the jejunum mucosa than those in the other groups (*p* < 0.01).

### 3.7. mRNA Expressions of Intestinal Tight Junction Protein and Inflammatory Cytokines of Suckling Piglets

Compared with the CON group, the piglets in the SM, SP, MP, and SMP groups showed significantly upregulated mRNA expressions of *CLDN-1* and *ZO-1* in the jejunal mucosa (*p* < 0.01, Figure 2A), and the piglets in the SMP group had significantly upregulated mRNA expressions of *OCLN* (*p* < 0.01). There was no significant difference in the mRNA expressions of *IL-6* among the five groups (*p* > 0.05, Figure 2B). However, the mRNA expressions of *TNF-α*, *IL-1β*, and *TLR4* of the jejunum were significantly downregulated in the SM, SP, MP, and SMP groups compared with those in the CON group (*p* < 0.01, Figure 2B,C), and the mRNA expressions of *NF-κB* in the jejunal mucosa were downregulated in the SM, SP, and SMP groups compared with those in the CON group (*p* < 0.01, Figure 2C).

### 3.8. Intestinal Microbial Flora in Colonic Digesta

The microbial flora in the colonic digesta was analyzed. As shown in Figure 3, a total of 222 OTUs were shared among the five treatment groups (Figure 3A). The piglets in the CON, SM, SP, MP, and SMP groups had 167, 317, 189, 164, and 205 unique OTUs, respectively (Figure 3A). However, there was no significant difference in the α-diversity (Shannon, Chao1, ACE, and Simpson indexes) of the colonic digesta in the suckling piglets among the five groups (*p* > 0.05, Figure 3B). The SM, SP, and MP groups resulted in significant changes in the beta diversity of the colonic microbiota, as shown by the NMDS based on the UniFrac distances (Figure 3C). Dietary supplementation with SMP was associated with increased relative abundances of *Faecalibacterium* at the genus level (Figure 3F, LDA score >2). Additionally, the piglets in the MP group were associated with increased relative abundances of *Collinsella*, while the piglets in the SM group were associated with increased relative abundances of *Catenbacterium*, *Coprococcus*, and *Bulleidia* at the genus level (Figure 3F, LDA score >4).

At the phylum level, the relative abundances of *Bacteroidetes* were increased in the SMP group (*p* < 0.05, Figure 4A), while the relative abundances of *Firmicute* in the SM, MP, and SMP groups were significantly lower than those of the CON group (*p* < 0.05, Figure 3B). At the genus level, the relative abundances of *Prevotella*, *Coprococcus*, and *Blautia* were increased in the SM group (*p* < 0.05, Figure 4C,F,H), and the relative abundances of *Blautia* and *Bulleidia* in the MP group were higher than those in the CON group (*p* < 0.05, Figure 4F,H). Compared with the piglets in the CON group, the piglets in the SMP group had increased relative abundances of *Faecalibacterium* (*p* < 0.05, Figure 4E).

## 4. Discussion

The ADFI of the sows and the ADGs of the piglets were considered the principal limitation factors for the growth performances of the neonatal pigs. In the present study, the dietary addition of SMP led to the highest ADG (236 g in the SMP group vs. 186 g in the CON group) and final BWs of the piglets (6.48 kg in the SMP group vs. 5.36 kg in the CON group), which agree with Jin et al. [26], who reported that the addition of fish oil could produce positive effects in the ADGs of piglets. Gebhardt et al. [27] found similar results: MCFAs improved the growth performance of nursery piglets by increasing the ADG, ADFI, and feed conversion ratio in a linear dose-dependent manner. Conversely, in some studies, researchers found no effect on the ADGs of piglets using 0.2% or even lower fatty acid products [27,28], which indicates that the ADGs were affected by the dietary fatty acid contents. It has been documented that SB has a distinct cheese flavor, which regulates appetite and food intake [10,29]. Our results demonstrated that the dietary addition of SM and SMP significantly increased the ADFI of the sows, whereas the dietary addition of SP and MP obtained the reverse results. Hanczakowska et al. [30] found similar results: a mixture of SCFAs and MCFAs produced better results on the ADFI. Additionally, Smit et al. [31] indicate that n-3 PUFAs have a higher energy density and boost overall energy intake, which allows sows to eat less. Thus, a mixture of SB and MCFAs might have potential additive effects to mitigate the negative effect of n-3 PUFAs on the ADFI.

In addition, diarrhea is the critical factor that causes the retardation of growth performances and increased mortality of piglets [32]. Feng et al. [13] and Lerner et al. [33] indicated that SB and MCFAs have the potential to replace antibiotics to control pathogenic bacteria, while n-3 PUFAs have a positive immunomodulatory effect in the gut [34,35]. In this study, the diarrhea incidence of the suckling piglets in the SM, SP, and SMP groups was lower than that in the CON group. Similar results were reported by Chen et al. [20]. The reason might be that SB can provide conditions for the growth of beneficial intestinal bacteria to reduce diarrhea incidence [19]. Intriguingly, Li et al. [36] indicate that organic acid combined with MCFAs showed a better reduction in the diarrhea incidence and growth-promoting effects that were comparable to those of antibiotics. It is noteworthy that supplementing the diets of the sows with SMP had the obvious and unexpected effect of decreasing the diarrhea incidence of the nursing piglets, which indicates that a blend of SB, MCFAS, and n-3 PUFAs may have a synergistic effect, although the mechanism still needs further study.

The composition and intake of the colostrum are crucial factors that affect the early weight gain, immune function, and survival of neonatal pigs with limited energy reserves [37]. Previous studies have shown a positive association between the content of fat in the milk and the piglet BW, and maternal fat supplementation could improve the piglet weaning weight [38,39]. Consistently, dietary supplementation with fish oil has increased the concentrations of fat in the colostrum of sows [40]. In this study, compared with the CON diet, sows fed the addition of SM, SP, MP, and SMP diets showed increased concentrations of fat and protein in the colostrum, which is in agreement with previous studies. In addition, the immunoglobulins in the colostrum are the only source of passive immunity for neonatal piglets. The colostral IgA and IgG concentrations are major factors that influence the passive immune protection. Jin et al. [26] indicated that dietary fish oil supplementation increased the IgG and IgM concentrations in the colostrum of suckling pigs, which improved their immune function. Similarly, a diet supplemented with SB increased the concentration of IgA in the colostrum [12]. Our results indicated that the concentrations of IgA and IgG in the colostrum of the SM, SP, and SMP groups were higher than those of the sows in the CON group. Furthermore, similar results were seen in the piglet plasma, which indicated that fatty acids played a positive role in regulating the immune status and providing health benefits. Additionally, the addition of SMP to the sows’ diets increased the IgA and IgM concentrations in the colostrum compared with those of the sows fed the MP diets, with no significant changes observed for the SM, SP, or SMP additions. SB and omega-3 PUFAs exert multiple beneficial effects, including immunomodulatory effects. He et al. [12] also observed that the IgA concentrations in the colostrum increased in SB-treated gilts. A possible explanation is that the blends of SB, MCFAs, and omega-3 PUFAs fed to the sows had additive effects. Collectively, dietary supplementation with fatty acids could enhance the growth performances of piglets by improving the colostrum composition.

The plasma biochemical parameters can reflect the nutritional status of the organism, and they can be influenced by changes in internal and external factors. The contents of TP and BUN are used as an index for protein utilization and metabolism. In this study, the piglets in the fatty acid-supplemented groups showed increased contents of plasma TP, which was partially due to the increment in the plasma globulin content. Meanwhile, the addition of fatty acids with different chain lengths to the sow diets increased the contents of BUN in the piglets, which indicated increased nitrogen metabolism. Moreover, previous studies on rats and pigs have shown the beneficial effects of fatty acids on lipid metabolism [41,42]. It is well documented that the TC and TG contents reflect the synthesis and metabolism of lipids in the organism and are associated with diseases linked to dyslipidemia [43]. Allyson et al. [44] reported that HDL cholesterol has a strong transport function that delivers cholesterol from peripheral cells to the liver cells. Similarly, Yu et al. [41] showed that the addition of SCFAs to the diet promotes lipid catabolism. In this study, the piglets in the SM, MP, and SMP groups showed increased HDL-cholesterol levels and lower TG and TC contents, which suggest that fatty acids might reduce cholesterol deposition in the blood.

GSH-Px, SOD, and CAT are considered to be important endogenous antioxidant enzymes that scavenge endogenous free radicals produced by the body, maintain the body’s oxidative balance, and play an important role in the oxidative and antioxidative status [45]. In this study, the piglets in the SM, SP, and MP groups exhibited increases in the activities of T-AOC CAT, GSH-Px, and T-SOD in the plasma. Famurewa et al. [46] also found that the dietary addition of coconut oil relieved oxidative stress in a dose-dependent manner by significantly increasing the antioxidant enzyme activities (SOD, CAT, and GSH-Px). Similarly, Nguyen et al. [47] reported the benefits of a diet rich in n-3 PUFAs in stimulating antioxidant enzyme activities to reduce excess ROS production. MDA is considered to be the main product of lipid oxidation, and it is a commonly used indicator of lipid peroxidation [47]. Li et al. [48] showed that the addition of MCFAs to the diet linearly reduced the plasma MDA concentration. Unexpectedly, in this study, the dietary addition of a combination of mixed MCFAs and n-3 PUFAs increased the plasma MDA levels. The discrepancies between studies might be due to the different doses of n-3 PUFAs and the lengths of time that the n-3 PUFAs were supplied. Diets rich in n-3 PUFAs may undergo peroxidation, which leads to free radical-dependent cellular damage, as evidenced by elevated plasma MDA levels [49,50]. These results indicate that the fatty acid alleviation in the oxidative stress statuses of piglets might be related to the improvement in cholesterol metabolism.

Tight junction proteins play an important role in intestinal barrier function. It has been reported that *OCLN* and *ZO-1*, which are tight junction proteins, are vital in the regulation of intestinal permeability [51]. Feng et al. [13] indicated that SB significantly increased the intestinal *ZO-1* and *OCLN* expressions in vivo and in vitro. A prior study has demonstrated that different fatty acid treatments were beneficial to the intestinal epithelial barrier integrity and intestinal barrier function [20]. In the current study, we observed that the piglets in the SM, SP, MP, and SMP groups had significantly upregulated mRNA expressions of *CLDN-1* and *ZO-1* in the jejunal mucosa, which contributed to the alleviation of diarrhea in the suckling piglets. The piglets in the SMP group showed higher mRNA expressions of *CLDN 1*, *ZO-1*, and *OCLN* than those in the other groups, which implies that these three fatty acids may have a synergistic effect in strengthening the intestinal barrier function. In addition, the intestinal mucosa morphology is an evaluation of the nutrient digestion and absorption ability, which has a direct impact on the nutrient usage efficiency [23]. It is well-known that SB can reduce some of the negative effects of the intestinal mucosal morphology by providing the preferred energy [10]. Keyser et al. [52] indicated that MCFA supplementation restored the villus height in postweaning piglets with LPS challenges. In addition, n-3 PUFAs may repair the gut damage induced by oxidative stress and enhance the intestinal morphology in piglets [34]. These combined findings indicate that fatty acid supplementation could slightly improve intestinal development by enhancing the barrier integrity and intestinal morphology.

Fatty acids play a major role in the inflammatory response of intestinal mucosa [25,53]. Many studies indicate that n-3 PUFAs can alleviate the inflammatory status in animals [17,35]. Carlson et al. indicated that medium-chain triglycerides reduced the mRNA expressions of *IL-6* and *TNF-α* in mice and alleviated the inflammatory response [54]. Similarly, Kuang et al. [55] demonstrated that the addition of mixing MCFAs with SCFAs to the base diet significantly reduced the mRNA expressions of *TNF-α* and *IL-1β*. In the present study, the mRNA expressions of the inflammatory factors (*TLR4, IL-1β, MγD88, TNF-α,* and *NF-κB*) in the jejunal mucosa were reduced, while the mRNA expression of the anti-inflammatory factor (*IL-10*) was increased. Researchers have hypothesized that fatty acids of different chain lengths likely attenuate the inflammatory response through the NF-κB and TLR4 signaling pathways [56,57]. Fatty acids inhibit inflammatory factors by regulating the MγD88-dependent route. In addition, the receptor for *TLR4* upregulated the *MγD88* expression, which can lead to the production and release of inflammatory factors, inducing an immune response in the intestinal mucosa [58,59,60]. Of note, the mRNA expressions of *IL-1β*, *TLR4*, and *TNF-α* in the SM, SP, MP, and SMP groups were lower than those of the CON group, while the mRNA expressions of *IL-10* had an opposite result in the SM, SP, and SMP groups. Butyric acid may be related to an inhibitor of a histone deacetylase and result in better anti-inflammatory effects [9].

Intestinal microbes play important roles in the host health and performance, and they can profoundly impact the host nutrient metabolism, intestinal development, and immunological functions [61]. Researchers have extensively demonstrated that fatty acids can modulate the abundance and composition of intestinal microbes. *Firmicutes*, *Bacteroidota*, *Proteobacteria*, and *Actinobacteriota* predominated in the colonic contents of the suckling piglets, which is consistent with previous studies [19,62]. Researchers have reported that *Bacteroidetes* was able to significantly reduce the diarrhea incidence [63]. In the present study, supplementing the sow diets with SMP significantly increased the relative abundance of *Bacteroidetes* in the colonic digesta of the suckling piglets, which may partly explain the reduction in the diarrhea incidence. At the genus level, the high abundance of bacteria provides an opportunity to understand how microbiota metabolites affect the host physiology. *Prevotella* belongs to *Bacteroidetes*, while *Faecalibacterium*, *Blautia*, *Bulleidia*, and *Coprococcus* belong to *Firmicutes*, in which *Prevotella* and *Coprococcus* are mostly involved in complex polysaccharide metabolism [64]. *Prevotella*, *Faecalibacterium*, *Blautia*, and *Coprococcus* can produce high levels of SCFAs, mainly including propionate, butyrate, and acetic acid [64,65,66]. Butyric acid has been identified as a major energy source for colonic epithelial cells [67]. In this study, we found that the piglets in the SM group had higher relative abundances of *Coprococcus* than those in the CON group, which indicates the potential for the increased intestinal availability of butyrate. Similarly, a reduced diarrhea incidence has been proven to be one of the strategies by which *Prevotella* improves the intestinal immunity and promotes animal growth [68,69]. Our results indicated that the relative abundances of *Prevotella* in the SM group were higher than those in the other groups, which partially agrees with Li et al., who observed that the combination of SCFAs and MCFAs increased the relative abundances of *Prevotella* [36]. We speculated that SB, along with MCFA supplementation, might modulate the gut microbiota composition and benefit the host’s health. Moreover, it has been reported that *Faecalibacterium* is an anti-inflammatory intestinal commensal microbe that can suppress the TLR4/NF-κB signaling pathway in intestinal epithelial cells. Importantly, we found that the relative abundances of *Faecalibacterium* were particularly increased in the colonic digesta of the piglets when the sows were fed the SMP diet, and it might play an anti-inflammatory role and could promote intestinal development.

## 5. Conclusions

In conclusion, our results indicate that diets supplemented with different combinations of SB, MCFAs, and omega-3 PUFAs during late gestation and lactation can efficiently improve the growth performance, immune function, antioxidant capability, and intestinal microbiota, as well as decrease the incidence of diarrhea, in suckling piglets. Additionally, dietary SMP supplementation had better effects on piglet intestinal health and probably through gut microorganism alterations. In the future, attention should be focused on the dosage of fatty acid additives to sow diets during lactation, the synergistic effect of multiple fatty acids, and the mechanism of the interactions between fatty acids and intestinal microorganisms.

## Figures and Tables

**Figure 1 animals-13-01093-f001:**
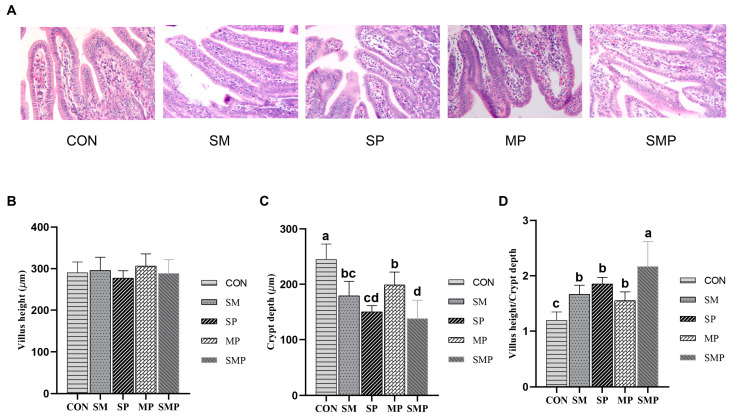
Effects of supplementing the diets with different combinations of SB, MCFAs, and omega−3 PUFA on the intestinal morphology in suckling piglets. Intestinal morphology of the jejunum was shown (**A**) among the five groups. Villous height (**B**), crypt depth (**C**), and villus height/crypt depth (**D**) in the jejunum (shooting magnification of 400×). Data are presented as means ± SEM (*n* = 6). ^a–d^ Values within a row with different superscripts differ significantly at *p* < 0.05.

**Figure 2 animals-13-01093-f002:**
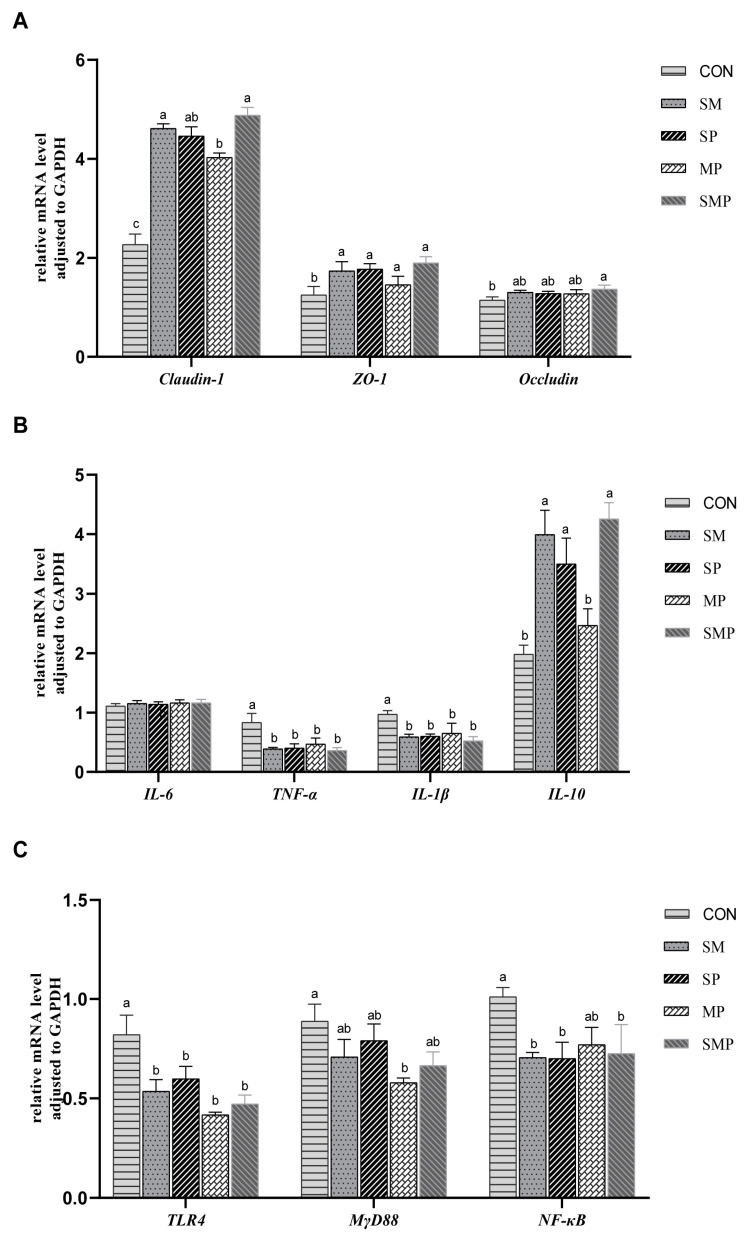
Effects of supplementing the diets with different combinations of SB, MCFAs, and omega−3 PUFA on the tight junction proteins and inflammatory cytokines in the jejunal mucosa of suckling piglets. The relative expression of *Claudin-1*, *ZO-1*, *Occludin*, *IL-1β*, *IL-6*, *IL-10*, *TNF-α*, *TLR-4*, *MγD88*, and *NF-κB* mRNA in the jejunal mucosa (**A**–**C**) were determined via real-time PCR. Data are expressed as the means ± SEM (*n* = 6). ^a–c^ Values within a row with different superscripts differ significantly at *p* < 0.05.

**Figure 3 animals-13-01093-f003:**
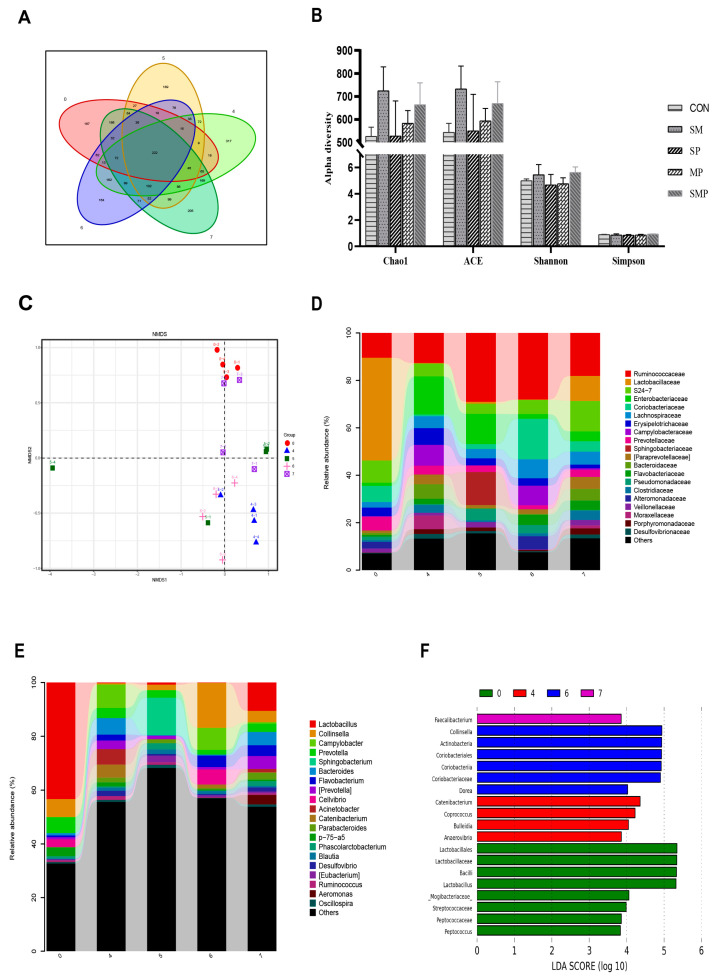
Effects of supplementing the diets with different combinations of SB, MCFAs, and omega−3 PUFA on microbial counts in the colonic digesta of suckling piglets. (**A**) Venn diagram shows the common and special OTUs distribution among the five groups. (**B**) Alpha diversity of the colonic digesta. (**C**) Nonmetric Multidimensional Scaling analysis (NMDS) on genus level based on UniFrac distances. The relative abundance of colonic microbiota at the phylum level (**D**) and the genus level (**E**). (**F**) The LEfSe analysis screened biomarkers of the microbial community. Data are expressed as the means ± SEM (*n* = 4). Note the 0, 4, 5, 6, and 7 groups in the legends indicate the CON, SM, SP, MP, and SMP groups, respectively.

**Figure 4 animals-13-01093-f004:**
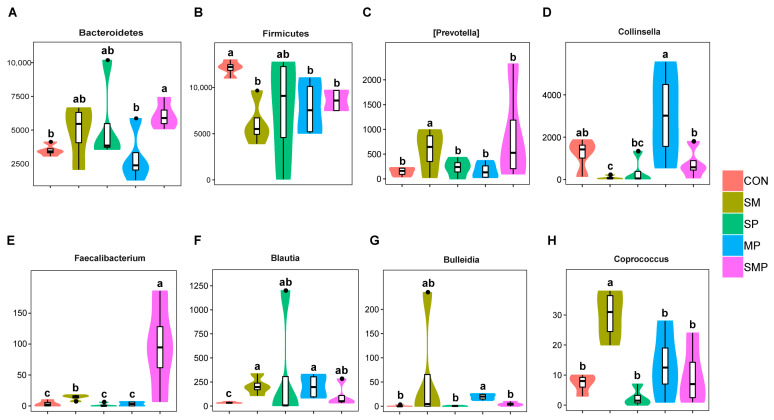
Effects of supplementing the diets with different combinations of SB, MCFAs, and omega−3 PUFA on specific bacteria in the colonic digesta of suckling piglets. Data are expressed as the means ± SEM (*n* = 4). ^a–c^ Values within a row with different superscripts differ significantly at *p* < 0.05. (**A**,**B**) Significantly different species of the colonic digesta at the phylum level. (**C**–**H**) Significantly different species of the colonic digesta at the genus level.

**Table 1 animals-13-01093-t001:** The formulations and chemical composition of the basal diet (as-fed basis).

Items	Stages	
	Late Gestation	Lactation
Ingredients	%	%
Corn	50.04	47.65
Barley	17.40	18.00
Soybean meal	17.20	19.00
Expanded soybean	6.00	6.00
Soybean oil	2.70	2.70
Fish meal	2.00	2.00
Limestone	1.60	1.60
CaHPO_4_	1.40	1.40
NaCl	0.40	0.40
Lys	0.26	0.25
Premix ^1^	1.00	1.00
Total	100.00	100.00
Composition		
DE ^2^, Mcal/kg	3.39	3.42
CP ^3^, %	15.40	15.90
EE ^3^, %	5.00	5.10
Ash ^3^, %	5.80	5.90
CF ^3^, %	3.90	3.50
Ca ^3^, %	1.07	1.20
Total P ^3^, %	0.63	0.74
Available P ^3^, %	0.50	0.59
Lys ^2^, %	1.14	1.17

^1^ The premix provided the following per kilogram of the diet: Cu 5 mg, I 0.15 mg, Fe 83 mg, Mn 20 mg Zn 128 mg, VA 13,400 IU, VD_3_ 2800 IU, choline chloride 1000 mg, VE 22.4 mg, VK_3_ 3 mg. The premix provided the following per kilogram of the diet for Lactation: Cu 15 mg, I 0.13 mg, Fe 82 mg, Mn 20 mg, Zn 128 mg, VA 10,000 IU, VD_3_ 2000 IU, VE 30 mg, and VK_3_ 1.5 mg. ^2^ DE and the levels of Lys are calculated values and others are measured values. DE and the levels of Lys were calculated according to NRC (2012) [21]. ^3^ Analyzed according to the procedures followed by standard AOAC (2000).

**Table 2 animals-13-01093-t002:** Primers for real-time PCR analysis.

Gene	Accession Number	Sequence (5′-3′)	Size (bp)	T_m_ Value
*ZO-1*	XM 003353439.1	Forward: GAGGATGGTCACACCGTGGT	169	56 °C
		Reverse: GGAGGATGCTGTTGTCTCGG		
*OCLN*	NM 001163647.2	Forward: TGGGTTAAAAACGTGTCGGC	105	60 °C
		Reverse: CACTTTCCCGTTGGACGAGT		
*CLDN 1*	NM 001161635.1	Forward: ACCCCAGTCAATGCCAGATA	155	54 °C
		Reverse: GGCGAAGGTTTTGGATAGG		
*IL-1β*	NM 9405217038	Forward: CAAGGAAGTGATGGCTAA	165	56 °C
		Reverse: ACCAAGGTCCAGGTTTT		
*IL-6*	NM 9405217033	Forward: TCAGTCCAGTCGCCTTCT	146	61 °C
		Reverse: CCTTTGGCATCTTCTTCC		
*IL-10*	NM 214041.1	Forward: CACTGCTCTATTGCCTGATCTC	136	58 °C
		Reverse: AAACTCTTCACTGGGCCGAAG		
*MyD88*	AB292176.1	Forward: GATGGTAGCGGTTGTCTCTGAT	148	60 °C
		Reverse: GATGCTGGGGAACTCTTTCTTC		
*TNF-α*	NM 214022.1	Forward: CCACGCTCTTCTGCCTACTGC	168	55 °C
		Reverse: GCTGTCCCTCGGCTTTGAC		
*NF-κB*	EU399817.1	Forward: CAGCCCTATCCCTTTACG	133	60 °C
		Reverse: GCCACAGCCTGAGCAA		
*TLR4*	NM_001113039	Forward: CATACAGAGCCGATGGTG	113	60 °C
		Reverse: CCTGCTGAGAAGGCGATA		
*GADPH*	AF017079.1	Forward: ACATCAAGAAGGTGGTGAAG	178	60 °C
		Reverse: ATTGTCGTACCAGGAAATGAG		

Abbreviations: ZO-1, zonula occludens-1; TLR4, toll-like receptor 4; MγD88, myeloid differentiation factor 88; NF-κB p65, nuclear factor-kappa B p65; TNF-α, tumor necrosis factor; IL-6, interleukin-6; GAPDH, glyceraldehyde-3-phosphate dehydrogenase.

**Table 3 animals-13-01093-t003:** Effects of supplementing the diets with different combinations of SB, MCFAs, and omega−3 PUFA on the reproductive performance of sows and the growth performance of their piglets.

Item	Treatments ^1^					SEM	*p*-Value
	CON	SM	SP	MP	SMP		
Reproductive performance of sows							
ADFI lactation, kg/d	6.46 ^c^	7.32 ^a^	6.96 ^bc^	6.37 ^c^	7.35 ^a^	0.21	<0.010
Total born	14.2	14.2	13.2	13.6	14.2	1.80	0.360
Born alive	13.4	13.4	12.4	12.2	12.8	1.02	0.210
Stillborn	0.80	0.80	0.80	1.40	1.40	0.54	0.821
Born alive rate, %	94.2	94.1	94.3	90.3	90.6	3.64	0.802
Initial BW, kg	1.44	1.30	1.57	1.53	1.52	0.07	0.074
Litter birth weight, kg	20.4	20.6	20.8	20.7	21.4	1.43	0.991
WEI, d	6.60 ^ab^	7.80 ^a^	5.20 ^b^	4.60 ^b^	4.00 ^b^	0.69	<0.010
Growth performance of piglets							
Survival rate, %	82.0	90.7	84.7	90.0	88.4	2.58	0.131
final BW, kg	5.36 ^c^	5.64 ^bc^	6.05 ^ab^	5.71 ^bc^	6.48 ^a^	1.33	<0.010
ADG, g/d	186 ^b^	207 ^ab^	213 ^ab^	199 ^b^	236 ^a^	8.33	<0.010
Diarrhea incidence, %	20.6 ^a^	14.7 ^b^	14.6 ^b^	17.6 ^ab^	13.8 ^b^	1.47	<0.010

^1^ CON, a basal diet; SM, the basal diet plus a blend of 1 g/kg SB and 7.75 g/kg MCFAs; SP, the basal diet plus a blend of 1 g/kg SB and 68.2 g/kg n-3 PUFA; MP, the basal diet plus a blend of 7.75 g/kg MCFA and 68.2 g/kg n-3 PUFA; SMP, the basal diet plus a blend of 1 g/kg SB, 7.75 g/kg MCFA, and 68.2 g/kg n-3 PUFA. Data are presented as means ± SEM (*n* = 6). ^a–c^ Values within a row with different superscripts differ significantly at *p* < 0.05.

**Table 4 animals-13-01093-t004:** Effects of supplementing the diets with different combinations of SB, MCFAs, and omega−3 PUFA on the colostrum composition of sows.

Item	Treatments ^1^					SEM	*p*-Value
	CON	SM	SP	MP	SMP		
Fat, %	3.67 ^b^	5.18 ^a^	5.08 ^a^	6.25 ^a^	4.92 ^a^	0.20	<0.010
Protein, %	12.3 ^c^	18.3 ^a^	16.4 ^ab^	16.6 ^ab^	14.9 ^b^	0.79	<0.010
Lactose, %	2.89 ^c^	3.19 ^bc^	3.69 ^a^	3.84 ^a^	3.25 ^bc^	0.16	<0.010
SNF, %	23.1 ^b^	27.7 ^a^	23.7 ^b^	28.3 ^a^	21.9 ^b^	1.18	<0.010
IgA, μg/mL	5.80 ^c^	9.04 ^ab^	9.43 ^ab^	7.16 ^bc^	9.98 ^a^	0.62	<0.010
IgG, μg/mL	45.1 ^b^	66.8 ^a^	62.5 ^a^	68.6 ^a^	71.5 ^a^	4.57	<0.010
IgM, μg/mL	45.1 ^b^	56.3 ^a^	64.0 ^a^	46.7 ^b^	60.6 ^a^	2.67	<0.010

^1^ CON, a basal diet; SM, the basal diet plus a blend of 1 g/kg SB and 7.75 g/kg MCFAs; SP, the basal diet plus a blend of 1 g/kg SB and 68.2 g/kg n-3 PUFA; MP, the basal diet plus a blend of 7.75 g/kg MCFAs and 68.2 g/kg n-3 PUFA; SMP, the basal diet plus a blend of 1 g/kg SB, 7.75 g/kg MCFAs, and 68.2 g/kg n-3 PUFA. Data are presented as means ± SEM (*n* = 6). ^a–c^ Values within a row with different superscripts differ significantly at *p* < 0.05. Abbreviations: SNF, solids nonfat; IgA, immunoglobulin A; IgG, immunoglobulin G; IgM, immunoglobulin M.

**Table 5 animals-13-01093-t005:** Effects of supplementing the diets with different combinations of SB, MCFAs, and omega−3 PUFA on the plasma biochemical index of suckling piglets.

Item	Treatments ^1^					SEM	*p*-Value
	CON	SM	SP	MP	SMP		
TP, g/L	56.1 ^c^	64.1 ^b^	72.6 ^a^	67.3 ^b^	71.8 ^a^	1.17	<0.010
Albumin, g/L	22.8 ^b^	21.7 ^b^	22.2 ^b^	30.0 ^a^	20.6 ^b^	0.78	<0.010
Globulin, g/L	33.2 ^c^	42.4 ^b^	50.4 ^a^	37.3 ^c^	51.2 ^a^	1.39	<0.010
Albumin/globulin	0.70 ^b^	0.52 ^c^	0.44 ^c^	0.81 ^a^	0.40 ^c^	0.04	<0.010
BUN, mmol/L	2.20 ^c^	2.88 ^ab^	3.28 ^a^	3.10 ^ab^	2.74 ^b^	0.14	<0.010
TG, mmol/L	0.64 ^b^	0.46 ^c^	0.67 ^a^	0.49 ^c^	0.46 ^c^	0.03	<0.010
FFA, μmol/L	200 ^b^	752 ^a^	869 ^a^	845 ^a^	887 ^a^	34.9	<0.010
TC, mmol/L	4.22 ^a^	2.07 ^bc^	3.94 ^a^	2.55 ^bc^	3.13 ^b^	0.26	<0.010
HDL, mmol/L	1.14 ^c^	2.02 ^a^	1.54 ^b^	1.75 ^ab^	1.58 ^b^	0.09	<0.010
IgA, μg/mL	8.85 ^b^	13.7 ^a^	10.7 ^b^	10.3 ^b^	10.1 ^b^	0.52	<0.010
IgG, μg/mL	43.5 ^c^	89.6 ^a^	91.2 ^a^	53.4 ^b^	64.0 ^b^	3.48	<0.010
IgM, μg/mL	84.7	71.9	73.3	85.6	88.4	4.30	0.062

^1^ CON, a basal diet; SM, the basal diet plus a blend of 1 g/kg SB and 7.75 g/kg MCFAs; SP, the basal diet plus a blend of 1 g/kg SB and 68.2 g/kg n-3 PUFA; MP, the basal diet plus a blend of 7.75 g/kg MCFAs and 68.2 g/kg n-3 PUFA; SMP, the basal diet plus a blend of 1 g/kg SB, 7.75 g/kg MCFAs, and 68.2 g/kg n-3 PUFA. Data are presented as means ± SEM (*n* = 6). ^a–c^ Values within a row with different superscripts differ significantly at *p* < 0.05. Abbreviations: TP, total protein; TG, triglycerides; BUN, blood urea nitrogen; TC, total cholesterol; HDL, high-density lipoprotein; FFA, free fatty acid.

**Table 6 animals-13-01093-t006:** Effects of supplementing the diets with different combinations of SB, MCFAs, and omega−3 PUFA on antioxidant capacity in the plasma of suckling piglets.

Item	Treatments ^1^					SEM	*p*-Value
	CON	SM	SP	MP	SMP		
T-AOC, U/mL	4.50 ^c^	5.76 ^b^	14.6 ^a^	12.0 ^a^	11.2 ^a^	0.14	<0.010
T-SOD, U/mL	96.6 ^b^	101 ^a^	104 ^a^	102 ^a^	107 ^a^	1.58	<0.010
CAT, U/mL	5.73 ^c^	10.7 ^a^	6.53 ^b^	8.31 ^b^	5.26 ^c^	0.24	<0.010
GSH-Px, umol/L	321 ^c^	584 ^b^	623 ^a^	659 ^a^	498 ^c^	12.4	<0.010
MDA, nmol/mL	2.81 ^b^	2.75 ^b^	3.01 ^b^	3.44 ^a^	2.96 ^b^	0.07	<0.010

^1^ CON, a basal diet; SM, the basal diet plus a blend of 1 g/kg SB and 7.75 g/kg MCFAs; SP, the basal diet plus a blend of 1 g/kg SB and 68.2 g/kg n-3 PUFA; MP, the basal diet plus a blend of 7.75 g/kg MCFAs and 68.2 g/kg n-3 PUFA; SMP, the basal diet plus a blend of 1 g/kg SB, 7.75 g/kg MCFAs, and 68.2 g/kg n-3 PUFA. Data are presented as mean ± SEM (*n* = 6). ^a–c^ Values within a row with different superscripts differ significantly at *p* < 0.05. Abbreviations: T-AOC, total antioxidant capacity; T-SOD, total superoxide dismutase; CAT, catalase; GSH-Px, glutathione peroxidase; MDA, malondialdehyde.

## Data Availability

The data presented in this study are available on request from the corresponding author.

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
