# Peer review of "Effects of Different Combinations of Sodium Butyrate, Medium-Chain Fatty Acids and Omega-3 Polyunsaturated Fatty Acids on the Reproductive Performance of Sows and Biochemical Parameters, Oxidative Status and Intestinal Health of Their Offspring"

_animals, 2023, doi:10.3390/ani13061093_

Round 1

Reviewer 1 Report

In this paper, the authors aimed to assess “Effects of different combinations of sodium butyrate, medium-chain fatty acids and omega-3 polyunsaturated fatty acids on the reproductive performance of sows and biochemical parameters, oxidative status and intestinal health of their offspring” by You et al,  which is an interesting and useful study.

Overall, this is an interesting paper with many analyses to support the findings. The subject is adequate with the overall journal's scope. However, I have some remarks.

1.      Line 90-92   Please provide the ethical approval number of the author's institution

2.      Line 99    Please delete “used”

3.      Line 159,180  please verify  plasma

Line  249-254  Please provide the statistical methods for diarrhea rate data

Author Response

Response to Reviewer 1

Dear editors and reviewers,

Thank you very much for your response letter and relevant comments, which are very helpful in our revision and improvement of the paper. We have read all comments carefully and revised the manuscript in detail according to the suggestions, and changes are marked up using the “Track Changes” function. In addition, we have checked the relevant literature again in the manuscript and have made revisions. We hope that the amended version of the paper is approved and look forward to hearing from you.

Best regards,

Caiyun You

Response to Reviewer 1

Reviewer(s)' Comments to Author:

Point 1: Line 90-92 Please provide the ethical approval number of the author's institution.

Response 1: Thanks for your suggestion, we have corrected the detailed description of the ethical approval number and revised the manuscript carefully. This information has been added in Line 96.

Point 2: Line 99 Please delete “used”.

Response 2: Based on the comment, the mistake and have been deleted, please see Line 103.

Point 3: Line 159,180 please verify “plasma”.

Response 3: Thanks for your suggestion. We checked the manuscript again. It has been verified as "plasma".

Point 4: Line 249-254 Please provide the statistical methods for diarrhea rate data.

Response 4: Thanks for your suggestion, we re-checked the data and revised the manuscript carefully. The diarrhea rate data did not satisfy the normal distribution. Therefore, diarrhea rate data assessments were translated using the arcsine square root transformation for subsequent statistics. This information has been added in Line 284-287.

Reviewer 2 Report

This manuscript about the effects of different combinations of sodium butyrate, medium-chain fatty acids and omega-3 polyunsaturated fatty acids on the reproductive performance of sows and biochemical parameters, oxidative status and intestinal health of their offspring, comes with relevance for the pig production context today, when the selection for highly prolific sows over the last decades is facing the challenges derived from it. Improving sow and piglet health under the high demanding pig industry is of great interest, and feed additives are on the spotlight for providing solutions. This study investigates combinations of 3 different feed additives for sows to provide better health to sows and piglets, including sodium butyrate, medium-chain fatty acids and omega-3 polyunsaturated fatty acids.

Line 22-23: “Meanwhile, the blends of SB, MCFAs, and omega-3 PUFA supplementation to a basal diet showed the greatest extent.” à the greatest extent of what? Please rephrase for clarification.

Line 49: “high-yielding lactating sows” à if referring to high milk production capacity, the use of yielding and lactating are redundant

Comment: Number of animals, n=6/treatment; did the authors perform a statistical power analysis of the experimental design?

Line 106-107: “Heaters and exhaust fans kept the room at a comfortable 20 to 25 °C.” à … at a comfortable temperature of 20 to 25 °C.

Lines 109-115: Diets and Experimental design

Comment: How were the doses for each of the additives chosen? Could you provide a justification?

Comment: In the manuscript the authors mention that the presented study shared the animals in the control group with the previously published Chen et al. 2019. More detailed explanation on the experimental design and the shared circumstances with Chen et al would be needed.

Did Chen et al. and the present study share experimental design and barn at the time of performance? If so, why did the authors decide to analyze the data of the treatment groups in Chen et al 2019, which include the supplementation of sows with the individual additives, separately from those presented in the submitted manuscript?

The present study investigates the interaction effects between additives, however comparison to their single administration as in Chen et al. is not visibly used for comparison and potential elucidation of synergistic effects [line 16] or incompatibilities. Did the authors consider a global analysis with all groups included?

Alternatively, the experiment presented in this study could have performed at a different time. If that were the case, perhaps a reconsideration of the statistical analysis and a better detailed description of the experimental design would be needed.

Line 169: “One colostrum sample (approximately 20 mL) diluted with 3 times purified water  was determined in duplicate for…” à …diluted 3 times with purified water…? please rephrase for clarification of the dilution.

Lines 221-224: consider adding annealing temperatures for each of the primers in table 2 to improve readability.

Lines 319-320: “However, the content of plasma MDA was increased in the MP group than that of the CON group…”à …was increased in the MP group compared to that of the… OR …was higher in the MP group than in the CON…

Figure 1: histology images of poor quality and lack of microscopic scale.

Figure 3: Legends indicate groups 0, 4, 5, 6, 7. Numbers are not declared as of which treatment do they correspond to, please indicate or correct the figures.

Results and Discussion: The results obtained in the treatment groups are compared individually to the control group. As previously mentioned, the authors mention the goal of investigating synergies between additives, however, they are not explored in depth in the manuscript, which could further improve the quality and interest of the research. 

Author Response

Response to Reviewers 2

Reviewer(s)' Comments to Author:

Point 1: Line 22-23: “Meanwhile, the blends of SB, MCFAs, and omega-3 PUFA supplementation to a basal diet showed the greatest extent.” the greatest extent of what? Please rephrase for clarification.

Response 1: Based on the comment, your helpful suggestions. It is very necessary to include an appropriate description, so we have revised the sentence accordingly, please see Line 24-25.

Point 2: Line 49: “high-yielding lactating sows” if referring to high milk production capacity, the use of yielding and lactating are redundant

Response 2: Based on the comment, it has changed high-yielding lactating sows to high-yielding sows, please see Line 52.

Point 3: Comment: Number of animals, n=6/treatment; did the authors perform a statistical power analysis of the experimental design?

Response 3: According to the comment, we did not do the power analysis before the animal experiment, however, based on previous studies (Harihara Iyer et al., 2012; Luo et al., 2009; Raj et al., 2017), we thought 6 replicates per treatment was enough. And based on your advice, we did the post doc power analysis. According to the post hoc power analysis for ADFI of sows and ADG of piglets, IgA in the colostrum and plasma in this study, statistical power we calculated were >0.90, so 6 pigs per treatment were enough to give sufficient statistical power (α < 0·05; β = 0·80). We have carefully focused on these suggestions and revised the manuscript accordingly. However, we do really need to increase the animal number in the future study. The power calculation was performed in this study and we have added a brief description in Statistical analysis part, please see Line 284-287.

Point 4: Line 106-107: “Heaters and exhaust fans kept the room at a comfortable 20 to 25 °C.” at a comfortable temperature of 20 to 25 °C.

Response 4: We have changed ‘at a comfortable 20 to 25 °C’ to ‘at a comfortable temperature of 20 to 25 °C, please see Line 112 .

Point 5: Lines 109-115: Diets and Experimental design.

Response 5: Thank you for your suggestion. We completely agree that it is essential for readers to have details on the diets and experimental design and have thus added them; please see Lines 121-124.

Point 6: Comment: How were the doses for each of the additives chosen? Could you provide a justification?

Response 6: Thank you for your advice; it is very important to select the dosage of additives for animal trials. The dosage of sodium butyrate, medium-chain fatty acids, and omega-3 poly-unsaturated fatty acids was chosen based on the company's recommended dosage and has been added in M & M part, please see Line 121.

Point 7: Comment: In the manuscript the authors mention that the presented study shared the animals in the control group with the previously published Chen et al. 2019. More detailed explanation on the experimental design and the shared circumstances with Chen et al would be needed.

Response 7: Thanks for your suggestion, we have corrected the detailed description of the experimental design and revised the manuscript carefully. This information has been added in Line 125-126.

Point 8: Did Chen et al. and the present study share experimental design and barn at the time of performance? If so, why did the authors decide to analyze the data of the treatment groups in Chen et al 2019, which include the supplementation of sows with the individual additives, separately from those presented in the submitted manuscript?

Response 8: That is a very interesting question deserving to be noticed. The present study did share the control group and barn with Chen et al. However, sows were not fed diets with fatty acids individual followed by diets with different combinations of fatty acids. Forty-eight sows were used in this trial and the previous trial. Forty-eight sows were randomly allocated to eight treatments (six replicate pens per treatment and one sow per replicate). Sows were fed a basal diet (control, CON), a basal diet supplemented with 1 g/kg of coated SB (SB), a basal diet supplemented with 7.75 g/kg of coated MCFAs (MCFA), a basal diet supplemented with 68.2 g/kg of coated n-3 PUFA (n-3 PUFA), a basal diet supplemented with 1 g/kg of coated SB and 7.75 g/kg of coated MCFAs (SM), a basal diet supplemented with 1 g/kg of coated SB and 68.2 g/kg of coated n-3 PUFA (SP), a basal diet supplemented with 7.75 g/kg of coated MCFAs and 68.2 g/kg of coated n-3 PUFA (MP), and a basal diet supplemented with 1 g/kg of coated SB, 7.75 g/kg of coated MCFAs and 68.2 g/kg coated n-3 PUFA (SMP). Therefore, we decided to share the control group and to provide the data in the submitted manuscript.

Point 9: The present study investigates the interaction effects between additives, however comparison to their single administration as in Chen et al. is not visibly used for comparison and potential elucidation of synergistic effects [line 16] or incompatibilities. Did the authors consider a global analysis with all groups included?

Response 9: Thanks for the suggestion. We did not analyze all global analysis with all groups to avoid the problem of duplicate data publication. However, it is very necessary to explore them in depth in the manuscript. We explore in depth whether the combination of fatty acids has an additive effect on the reproductive performance of the sow and the growth performance of the offspring in the discussion part. We speculated that the combination of fatty acids would have additive effects on reproductive performance of sows and growth performance of their offspring. Thus, we specifically focused on the ADFI of sows, the composition of colostrum, and intestinal microbiota of piglets. 1) The ADFI of sows. Our results demonstrated that the dietary addition of SM and SMP significantly increased the ADFI of sows, whereas the dietary addition of SP and MP obtained the reverse results. Hanczakowska et al. [30] found similar results that a mixture of SCFA and MCFAs pro-duced better results on the ADFI. Additionally, Smit et al. [31] indicated that n-3 PUFA has a higher energy density and boosts overall energy intake allowing the sow from eating less. Thus, a mixture of Sb and MCFAs might have potential additive effects, to mitigate the negative effect of n-3 PUFA on the ADFI. 2) The composition of colostrum. The addition of SMP to sows' diets increased IgA and IgM concentrations in the colostrum compared to sows fed the MP diets, with no significant changes observed in the SM, SP, and SMP addition. It is widely established that SB and omega-3 PUFA exert multiple beneficial effects including immunomodulatory effect. He et al. [12] also observed that IgA concentrations in the colostrum increased in SB-treated gilts. A possible explanation was that blends of SB, MCFAs and omega-3 PUFA fed sows had additive effects. 3) Intestinal microbes. It has been reported that Faecalibacterium is an anti-inflammatory intestinal commensal microbe, which can suppress TLR4/NF-κB signaling pathway in the intestinal epithelial cells. Importantly, we found that the relative abundances of Faecalibacterium were particularly increased in the colonic digesta of piglets when sows were fed the SMP diet, which may might play an anti-inflammatory role in and could promoting promote intestinal development and benefit the host health. This information has been revised in Line 461-463, 472-475, 497-504, 602-608.

Point 10: Alternatively, the experiment presented in this study could have performed at a different time. If that were the case, perhaps a reconsideration of the statistical analysis and a better detailed description of the experimental design would be needed.

Response 10: The experiment presented in this study was conducted at the same time as the previous experiment (Chen et al.). It is noticed that the sows in this trial and Chen's sows were kept in the same barn. we corrected the section and have added detailed description of the experimental design in M & M part, please see Line 108-110 and 121-124.

Point 11: Line 169: “One colostrum sample (approximately 20 mL) diluted with 3 times purified water was determined in duplicate for…”…diluted 3 times with purified water…? please rephrase for clarification of the dilution.

Response 11: According to the comment, it has changed “diluted with 3 times purified water” to “diluted 3 times with purified water”, please see Line 183-184.

Point 13: Lines 221-224: consider adding annealing temperatures for each of the primers in table 2 to improve readability.

Response 13: Thank you for your valuable suggestions. We have added annealing temperatures for each of the primers in Table 2.

Point 14: Lines 319-320: “However, the content of plasma MDA was increased in the MP group than that of the CON group…”…was increased in the MP group compared to that of the… OR …was higher in the MP group than in the CON…

Response 14: We have corrected it, please see Line 361-362 .

Point 15: Figure 1: histology images of poor quality and lack of microscopic scale.

Response 15: The lack of microscale is very regretful owing to the lack of pixel conversion ratio. the shooting multiplier has been added and the image has been updated, please see Line 388. In addition, the lack of clarity of the images might be due to the instrument being outdated.

Point 16: Figure 3: Legends indicate groups 0, 4, 5, 6, 7. Numbers are not declared as of which treatment do they correspond to, please indicate or correct the figures.

Response 16: We have revised it based on your suggestion. Groups the 0, 4, 5, 6 and 7 in the legend represent the CON, SM, SP, MP and SMP groups respectively, please see Line 437-438.

Point 17: Results and Discussion: The results obtained in the treatment groups are compared individually to the control group. As previously mentioned, the authors mention the goal of investigating synergies between additives, however, they are not explored in depth in the manuscript, which could further improve the quality and interest of the research.

Response 17: According to the comment, it would like to clarify that our original goal was to investigate whether the combination of fatty acids had additive effects on reproductive performance of sows and growth performance of their offspring, rather than synergistic effects. We have revised this, please see Line 16 and 85.
